# FoleyCrafter: Bring Silent Videos to Life with Lifelike and Synchronized Sounds

## Abstract

We study Neural Foley, the automatic generation of high-quality sound effects synchronizing with videos, enabling an immersive audio-visual experience. Despite its wide range of applications, existing approaches encounter limitations when it comes to simultaneously synthesizing high-quality and video-aligned (*i.e.,* semantic relevant and temporal synchronized) sounds. To overcome these limitations, we propose FoleyCrafter, a novel framework that leverages a pre-trained text-to-audio model to ensure high-quality audio generation. FoleyCrafter comprises two key components: a semantic adapter for semantic alignment and a temporal adapter for precise audio-video synchronization. The semantic adapter utilizes parallel cross-attention layers to condition audio generation on video features, producing realistic sound effects that are semantically relevant to the visual content. Meanwhile, the temporal adapter estimates time-varying signals from the videos and subsequently synchronizes audio generation with those estimates, leading to enhanced temporal alignment between audio and video. One notable advantage of FoleyCrafter is its compatibility with text prompts, enabling the use of text descriptions to achieve controllable and diverse video-to-audio generation according to user intents. We conduct extensive quantitative and qualitative experiments on standard benchmarks to verify the effectiveness of FoleyCrafter. Models and codes will be available.

## 1 Introduction

Foley, a key element in film and video post-production, adds realistic and synchronized sound effects to silent media (contributors, 2024). These sound effects are the unsung heroes of cinema and gaming, enhancing realism, impact, and emotional depth for an immersive audiovisual experience. Traditionally, skilled Foley artists painstakingly create, record, and process sound effects in specialized studios, making it a labor-intensive and time-consuming process (Ament, 2014). Despite advancements in recent video-to-audio generation, achieving Neural Foley, which requires synthesizing high-quality, video-aligned sounds that are semantically related and temporally synchronized with the videos, remains challenging (Luo et al., 2023).

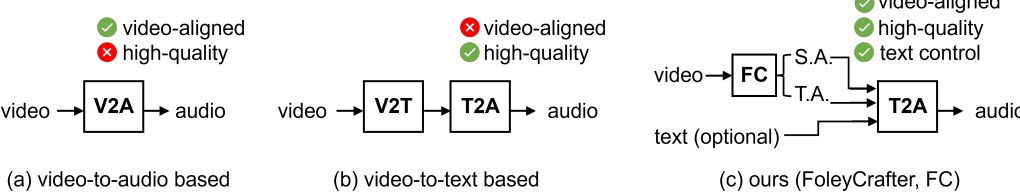

Figure 1: (a) (Video-to-audio) V2A methods struggle with audio quality due to noisy training data, while (b) video-to-text (V2T) methods encounter difficulties in producing synchronized sounds. Our model FoleyCrafter, integrates a learnable module into a pre-trained Text-to-Audio (T2A) model to ensure audio quality while enhancing video-audio alignment with the supervision of audios.

State-of-the-art approaches for Neural Foley in video-to-audio generation can be categorized into two main groups, as illustrated in Figure 1. The first group involves training a video-to-audio gen-

erative model on a large-scale paired audio-video dataset (Chen et al., 2020a; Iashin and Rahtu, 2021; Luo et al., 2023; Sheffer and Adi, 2023). However, the audio quality of such datasets crawled from the Internet can be subpar, with issues like noise and complex environmental sounds recorded in the wild, which hinder the production of high-quality sounds (Wang et al., 2024a; Xie et al., 2024b). To address this, the second group of approaches (Figure 1-(b)) adopts a two-stage process. They first translate video into text using video captioning or embedding mapping techniques and then employ a pre-trained text-to-audio model (Wang et al., 2024a; Xie et al., 2024b; Xing et al., 2024). Leveraging the well-trained text-to-audio generator, these methods achieve impressive sound quality. Nonetheless, effectively bridging the gap between video and text while preserving nuanced details is challenging. As a result, these methods often produce unsynchronized sounds due to the suboptimal translated text conditions.

To achieve both high-quality and video-aligned sound generation, we present FoleyCrafter, which breathes life into silent videos with realistic and synchronized sound effects. As depicted in Figure 1-(c), the core of FoleyCrafter is an innovative pluggable module that can be integrated with a pre-trained text-to-audio (T2A) model, optimized with the supervision of audios. Specifically, FoleyCrafter comprises two main components: a semantic adapter for semantic alignment and a temporal adapter for temporal synchronization. The semantic adapter introduces parallel cross-attention layers into the backbone of the T2A model. It takes as input the extracted video features, allowing FoleyCrafter to generate audio conditioned on the video without relying on explicit text. The temporal adapter, on the other hand, is engineered to refine temporal synchronization. The temporal adapter has two keysteps: figuring out time-varying signals from videos and matching them to synchronize audio generation. First, we study two ways to find audio signals in video frames. One way uses labels to detect when sounds start, and the other method uses energy maps without needing labels (Du et al., 2023). Then, we make the audio features line up with the video by matching them to these found audio signals. Such a design results in an enhanced video-synchronized audio generation. During training, we train the semantic adapter and temporal adapter with video-audio correspondent data, while fixing the text-to-audio base model to preserve its established audio generation quality. After training, FoleyCrafter can generate high-quality sounds for videos with semantic and temporal alignment in a flexible and controllable way.

We conduct extensive experiments to evaluate the performance of FoleyCrafter in terms of audio quality and video alignment, both semantically and temporally. Our experiments include quantitative analysis, qualitative comparison, and user studies, all of which demonstrate that FoleyCrafter has achieved state-of-the-art results. Additionally, we have showcased the controllability of FoleyCrafter through text prompts, allowing for a more fine-grained and versatile application of the model. Our main contributions can be summarized as follows:

- We present a novel Neural Foley framework that generates high-quality, video-aligned sound effects for silent videos, while also offering fine-grained control through text prompts.

- To ensure both semantic and temporal alignment, we design a semantic adapter and a temporal adapter, significantly improving video alignment.

- We validate the effectiveness of FoleyCrafter through extensive experiments, including quantitative and qualitative analyses. Our results show that FoleyCrafter achieves state-of-the-art performance on commonly used benchmarks.

## 2 RELATED WORK

**Diffusion-based Audio Generation.** Latent diffusion models have significantly advanced audio generation (Liu et al., 2023a;b; Rombach et al., 2022). AudioLDM pioneers open-domain text-to-audio generation using a latent diffusion model (Liu et al., 2023a;b). Tango improves text-to-audio generation with an instruction-tuned LLM FLAN-T5 (Chung et al., 2024a) as the text encoder (Ghosal et al., 2023). Make-an-Audio tackles complex audio modeling using spectrogram autoencoders instead of waveforms (Huang et al., 2023). Xue et al. conduct comprehensive ablation studies to explore effective designs and set a new state-of-the-art with the proposed Auffusion (Xue et al., 2024). Moreover, some works (Guo et al., 2024; Xie et al., 2024a; Chung et al., 2024b; Comunità et al., 2024; Jeong et al., 2024) have further studied the conditional generation with temporal order condition, promoting the controllability of these diffusion models. In this paper, we introduce Fol-

eyCrafter, a module that extends state-of-the-art text-to-audio generators to support video-to-audio generation while preserving the original text-to-audio controllability.

**Video-to-Audio Generation.** Foley artistry is a crucial audio technique that enhances the viewer's auditory experience by creating and recording realistic sound effects that synchronize with visual content (contributors, 2024). Early Neural Foley models mainly focus on generating sounds tailored to a specific genre or a narrow spectrum of visual cues, underscoring the potential of deep learning to innovate sound effect creation for videos (Chen et al., 2018; 2020b; Owens et al., 2016; Zhou et al., 2018). Despite recent advancements in large-scale generative models (Huang et al., 2023; Liu et al., 2023a), generating high-quality and visually synchronized sounds for open-domain videos remains a challenge (Dong et al., 2023; Du et al., 2023; Luo et al., 2023; Mo et al., 2024; Tang et al., 2024; Wang et al., 2024a; Comunità et al., 2024; Pascual et al., 2024; Wang et al., 2024b; Su et al., 2024).

State-of-the-art video-to-audio approaches can be categorized into two groups. The first group focuses on training video-to-audio generators from scratch (Iashin and Rahtu, 2021; Luo et al., 2023; Sheffer and Adi, 2023). Specifically, SpecVQGAN (Iashin and Rahtu, 2021) employs a cross-modal Transformer (Vaswani et al., 2017) to auto-regressively generate sounds from video tokens. Im2Wav (Sheffer and Adi, 2023) conditions an autoregressive audio token generation model using CLIP features, while Diff-Foley (Luo et al., 2023) improves semantic and temporal alignment through contrastive pre-training on aligned video-audio data. However, these methods are limited by the availability of high-quality paired video-audio datasets. An alternative approach is to utilize text-to-audio generators for video Foley. Xing et al. (2024) introduce an optimization-based method with ImageBind (Girdhar et al., 2023) for video-audio alignment, while SonicVisionLM (Xie et al., 2024b) generates video captions for text-to-audio synthesis. Wang et al. note the limitations of caption-based methods and propose V2A-Mapper to translate visual embeddings to text embedding space (Wang et al., 2024a). Nevertheless, effectively bridging the gap between video and text while preserving fine-grained temporal cues remains a significant challenge. In contrast, we introduce FoleyCrafter, integrating a learnable module into text-to-audio models with end-to-end training, enabling a high-quality, video synchronized and high-controllable Foley.

## 3 APPROACH

In this section, we introduce the framework of FoleyCrafter. We introduce related preliminaries about Audio Latent Diffusion Models (ALDMs) (Liu et al., 2023a;b) and conditioning mechanisms in Section 3.1. We then delve into the key components of FoleyCrafter in Section 3.2. The semantic adapter generates audio based on visual cues and text prompts, while the temporal adapter improves temporal synchronization with the video. We also outline the training process for each component and explain how FoleyCrafter can be used to generate foley for videos in Section 3.3.

### 3.1 PRELIMINARIES

**Audio Latent Diffusion Model.** The latent diffusion model (LDM) has achieved remarkable advancements in text-to-audio generation, as demonstrated by recent studies (Ghosal et al., 2023; Liu et al., 2023a;b; Xue et al., 2024). In this model, the audio waveform is initially transformed into a mel-spectrogram representation. Subsequently, a variational autoencoder (VAE) encodes the mel-spectrogram into a latent representation denoted as $z$. The LDM's UNet is trained to generate $z$ by denoising normally distributed noise $\epsilon$. The predicted latent $z$ is then reconstructed by the VAE into a mel-spectrogram, which is finally transformed into a waveform using a vocoder.

A latent diffusion model consists of two main processes: the diffusion process and the denoising process. In the diffusion process, a clean latent representation $z$ undergoes step-by-step noise addition until it reaches an independently and identically distributed noise. It can be denoted as,

$$z_t = \sqrt{\bar{\alpha}_t} z_0 + \sqrt{1 - \bar{\alpha}_t}\epsilon, \epsilon \sim \mathcal{N}(0, I) \tag{1}$$

where $\bar{\alpha}_t$ is the noise strength at $t$ timestep. The UNet is trained to estimate the added noise at a given timestep $t$ using the following optimization objective:

$$\mathcal{L} = \mathbb{E}_{x, \epsilon \sim N(0,1), t, c}\left[\|\epsilon - \epsilon_\theta(z_t, t, c)\|\right] \tag{2}$$

where $x$ represents the mel-spectrogram in the ALDM, $z_t$ corresponds to the latent representation of the mel-spectrogram at timestep $t$, and $c$ denotes the condition information.

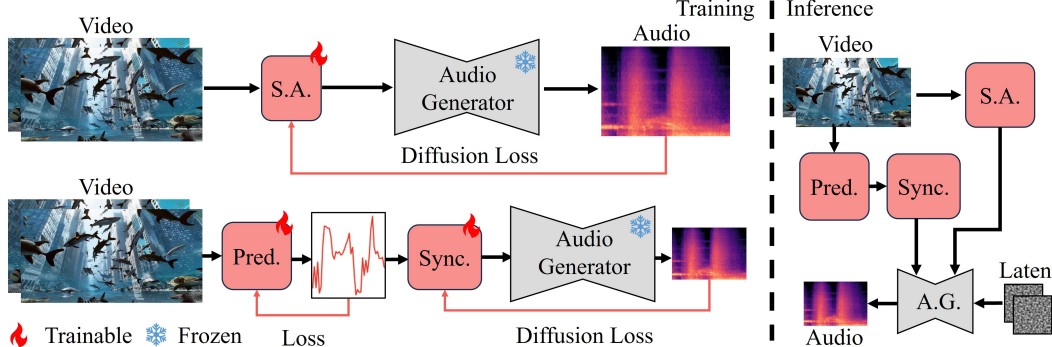

Figure 2: **The overview of FoleyCrafter.** FoleyCrafter is built upon a pre-trained text-to-audio (T2A) generator, ensuring high-quality audio synthesis. It comprises two main components: the semantic adapter (S.A.) and the temporal adapter. temporal adapter first predicts the time-varying signal from the video content (denoted as 'Pred.'), and then synchronize the audio with these estimated signals (denoted as 'Sync.'). Both the semantic adapter and the temporal adapter are trainable modules that take videos as input to synthesize audio, with audio supervision for optimization. The T2A model remains fixed to maintain its established capability for high-quality audio synthesis.

**Conditioning Mechanisms.** There are two kinds of condition mechanisms mainly used in ALDM, *i.e.,* MLP-based mechanism (Ghosal et al., 2023; Liu et al., 2023a) and cross-attention-based mechanism (Xue et al., 2024; Liu et al., 2023b). In the MLP-based mechanism, the time step is mapped to a one-dimensional embedding and concatenated with the text embedding as the conditioning information. This one-dimensional condition vector is then merged with the UNet's feature map through MLP layers. In contrast, the cross-attention-based mechanism utilizes cross-attention in each block of the UNet. This mechanism demonstrates improved alignment with conditions and allows for more flexible and fine-grained controllable generation. It has been widely adopted in recent works (Liu et al., 2023b; Xue et al., 2024). The cross-attention mechanism can be represented as follows:

$$Attention(Q, K, V) = softmax\left(\frac{QK^T}{\sqrt{d}}\right) \cdot V, \tag{3}$$

$$\text{where } Q = W_Q \cdot \varphi(z_t), \quad K = W_K \cdot \tau(c), \quad V = W_V \cdot \tau(c), \tag{4}$$

where $\varphi$ denote the flattening operation, $\tau$ is the condition encoder and $W_Q, W_K$ and $W_V$ is learnable projection matrices. In this study, we adopt a cross-attention mechanism to integrate textual and visual cues, aligning with recent state-of-the-art ALDMs (Liu et al., 2023b; Xue et al., 2024).

## 3.2 FOLEYCRAFTER

The FoleyCrafter comprises two core components: a semantic adapter for semantic alignment and a temporal adapter for temporal alignment. As illustrated in Figure 2, FoleyCrafter is a modular system that leverages a pre-trained text-to-audio (T2A) model (Freesound Project, 2024; Xue et al., 2024). This architecture enables FoleyCrafter to generate audio that is synchronized with videos, ensuring both high-quality and varied audio output. For our audio generation, we utilize Auffusion Xue et al. (2024) in our implementation. During training, only the two adapters are trainable, optimizing with the supervision of ground truth audio, while the weights of the T2A model remain fixed. In the following sections, we provide more details of each component.

### 3.2.1 SEMANTIC ADAPTER

To efficiently extract semantic features from the input video and incorporate them into the pre-trained text-to-audio generator, we employ a visual encoder along with decoupled parallel cross-attention layers. We demonstrate the overview of the semantic adapter in Figure 3.

**Visual Encoder.** The CLIP encoder has demonstrated its effectiveness as a semantic extractor for visual information (Radford et al., 2021). In our approach, we follow previous works (Rombach et al., 2022; Ye et al., 2023) and extract visual embeddings from each frame of the input video using

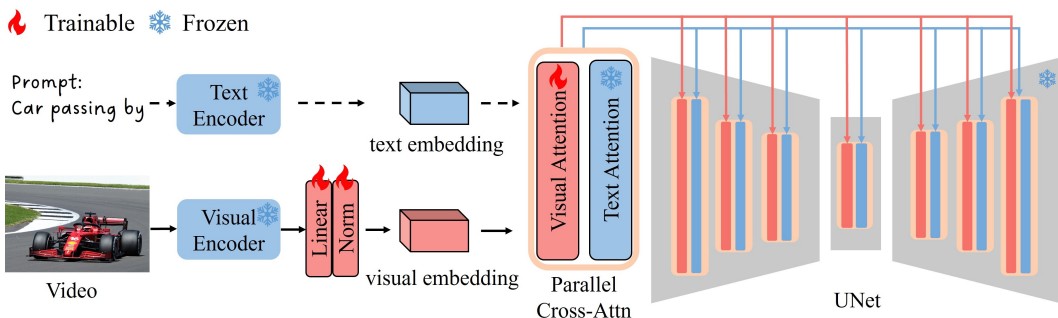

Figure 3: **The overview of semantic adapter**. Semantic adapter employs a pre-trained visual encoder with several learnable layers to extract video embeddings that align better with the text-to-audio generator. Then, it integrates trainable visual-cross attention mechanisms alongside text-based ones, ensuring semantic alignment with the video without compromising text-to-audio generation.

the CLIP image encoder. To align these embeddings with the text-to-audio generator, we employ several learnable projection layers. This process can be expressed as:

$$V_{emb} = MLP(AvgPooling(\tau_{vis}(v))). \tag{5}$$

Here, $v$ represents the input video, $\tau_{vis}$ denotes the CLIP image encoder, and $AvgPooling$ refers to the average pooling of the extracted CLIP features across frames.

**Semantic Adapter.** To incorporate the extracted video embedding into the pre-trained text-to-audio generator without compromising its original functionality, we introduce visual-conditioned cross-attention layers alongside the existing text-conditioned cross-attention layers. In this approach, visual and text embeddings are separately fed into their corresponding cross-attention layers. The outputs of the new and original cross-attention layers are then combined using a weight parameter, $\lambda$. The parallel cross-attention can be denoted as:

$$Attention(Q, K, V) = softmax(\frac{QK_{txt}^T}{\sqrt{d}}) \cdot V_{txt} + \lambda \cdot softmax(\frac{QK_{vis}^T}{\sqrt{d}}) \cdot V_{vis}, \tag{6}$$

$$\text{where } K_{txt} = W_K^{txt} \cdot T_{emb}, V_{txt} = W_V^{txt} \cdot T_{emb}, \tag{7}$$

$$K_{vis} = W_K^{vis} \cdot V_{emb}, V_{vis} = W_V^{vis} \cdot V_{emb}, \tag{8}$$

where $T_{emb}$ and $V_{emb}$ represent the extracted text embeddings and video embeddings, respectively. $W_K^{txt}$ and $W_V^{txt}$ correspond to the pre-trained projection layers in the text-conditioned cross-attention layers, which remain fixed during training. On the other hand, $W_K^{vis}$ and $W_V^{vis}$ are newly introduced learnable projection layers used to map the visual embedding to a space that aligns better with the condition space of the pre-trained text-to-audio generator.

During the training of the semantic adapter, we initialize the vision-conditioned cross-attention layers from the text-conditioned ones. As shown in Figure 3, we train the newly added projection layers after the visual encoder and the vision-conditioned cross-attention layers using ground truth audio as supervision. Meanwhile, we keep the text encoder and the text-to-audio generator fixed. The optimization objective can be expressed as:

$$\mathcal{L} = \mathbb{E}_{x,\epsilon \sim N(0,1),t,c} \left[ \|\epsilon - \epsilon_\theta(z_t, t, T_{emb}, V_{emb})\| \right]. \tag{9}$$

We noticed a related work, IP-Adapter, which is developed to inject image conditions into a pre-trained text-to-image diffusion model (Ye et al., 2023). However, it remains less explored in studying injecting a third modality (*i.e.,* video in our work) into a pre-trained text-to-audio diffusion model. We surprisingly find that our proposed semantic adapter can effectively extract meaningful semantic features from video frames and inject these features into audio features without compromising audio generation quality. To effectively capture visual cues for audio generation while retaining the capability of combining text prompts for more controllable video-to-audio generation, we randomly drop the text condition during training in the majority of cases (approximately 90%).

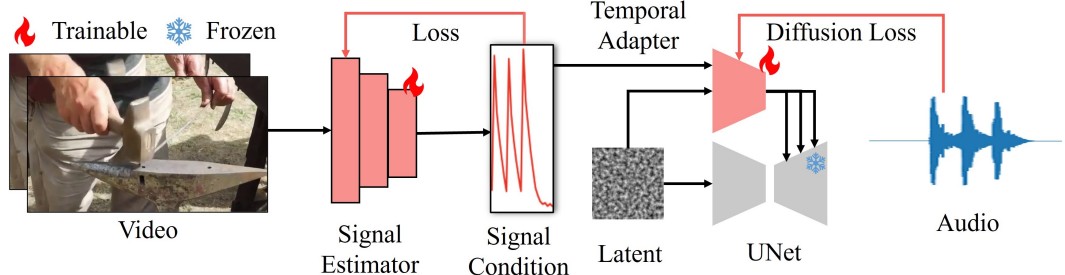

Figure 4: **The overview of the temporal adapter.** To enhance the temporal synchronization, temporal adapter takes a two step approach. Firstly, it predicts the audio signals from the visual cues, and then it utilizes the predicted signals to synchronize the audio with these estimated signals.

### 3.2.2 TEMPORAL ADAPTER

We observed that the semantic adapter captures video-level alignment without precise temporal synchronization for each frame. To address this limitation, we develop a temporal adapter to enhance the temporal synchronization. As shown in Figure 4, Overall, there are two key steps: first, estimating the time-varying signal from the video content, and second, synchronizing the audio with these estimated signals. This approach ensures that the generated audio features are precisely matched to the audio cues extracted from the video, resulting in improved temporal synchronization. We introduce more details of each step below.

**Audio Signal Estimation from Videos.** We study two distinct approaches for temporal estimation from video content: one that relies on timestamps and another that focuses on energy levels. In the timestamp-based method, a binary vector is used, where '1' indicates the presence of sound effects and '0' signifies silence. This type of timestamp signal has been demonstrated to effectively manage audio generation in a time-sensitive manner, as shown in Xie et al. (2024b); Comunità et al. (2024). To this end, we have developed an estimator designed to extract features from video inputs to forecast the presence of sound at specific timestamps. This timestamp-based estimator is trained using binary cross-entropy loss (Xie et al., 2024b), which is expressed as:

$$\mathcal{L}_{BCE}(y, \hat{y}) = -\frac{1}{N} \sum_{i=1}^{N} \left( y_i \log(\hat{y}_i) + (1 - y_i) \log(1 - \hat{y}_i) \right). \tag{10}$$

where $N$ represents the number of samples, $y$ denotes the ground truth, and $\hat{y}$ is the prediction.

While timestamp-based estimators excel at extracting audio-related information from videos, they often depend on the labor-intensive process of timestamp labeling. This reliance can restrict the quality and applicability of the estimator. To address this, we also investigate an alternative approach: energy map estimation. In this method, an energy map is derived from the mel-spectrum of the audio using a rule-based technique. The energy map effectively captures temporal audio characteristics like sustain and release, eliminating the need for manual labeling efforts (Jeong et al., 2024; Du et al., 2023). We train this energy map estimator using normalized energy values and mean squared error loss, which is formulated as follows:

$$\mathcal{L}_{MSE}(y, \hat{y}) = -\frac{1}{N} \sum_{i=1}^{N} ||y_i - \hat{y}_i||_2 \tag{11}$$

We employ a ResNet (2+1)-D18 convolutional network (Tran et al., 2018) as the temporal estimator. After training, the estimator can extract audio-related temporal information (i.e., timestamp or energy) from videos for further audio synchronization.

**Synchronizing Audios with Estimates.** Once we've extracted audio signals from videos, we implement a ControlNet (Zhang et al., 2023a) to serve as our synchronization module. This module aligns the audio features in the T2A model with the extracted estimates. Specifically, these estimated signals are interpolated to match the length of the audio latent, allowing them to act as a condition for the synchronization module. The synchronization module adopts the same architecture as the

UNet encoder in the T2A model. The output from this module is then added as a residual to the output of the original UNet to achieve synchronization. During the training phase, we focus on training the replicated UNet blocks, using the same optimization goal as the diffusion model.

### 3.3 IMPLEMENTATION DETAILS

For the semantic adapter, we follow Ye et al. (2023) to use a linear projection for clip visual embedding to better align with text representation and expand the embedding length to four. Then we modify all the cross-attention to parallel cross-attention for visual conditions. We train semantic adapter on the VGGSound (Chen et al., 2020a) for 164 epochs with a batch size of 128 on 8 NVIDIA A100 GPUs. For the temporal adapter, we train the predictor and temporal adapter separately. The predictor is trained on the subset of the VGGSound (Chen et al., 2020a) i.e. AVSync15 (Zhang et al., 2024), which has a higher audio-visual relevance. The temporal adapter is also trained on the VGGSound (Chen et al., 2020a) for 80 epochs. Note that the energy signal can be derived from the mel-spectrum, whereas timestamps require manual annotation. This reduces data labeling costs and provides us with more available data. After training, the two components in temporal adaptercan be combined together for inference and evaluation.

## 4 EXPERIMENTS

### 4.1 EXPERIMENTAL SETTINGS

**Baselines.** We conducted comprehensive evaluations of FoleyCrafter by comparing it with state-of-the-art approaches, namely SpecVQGAN (Iashin and Rahtu, 2021), Diff-Foley (Luo et al., 2023), V2A-Mapper (Wang et al., 2024a), Seeing-and-hearing (Xing et al., 2024) and SonicVisionLM (Xie et al., 2024b). Both quantitative and qualitative comparisons were employed. SpecVQGAN generates audio tokens autoregressively based on extracted video tokens. Diff-Foley utilizes contrastive learning for synchronized video-to-audio synthesis with its CAVP audio and visual encoder. V2A-Mapper translates visual CLIP embeddings to CLAP space, enabling video-aligned audio generation using a pre-trained text-to-audio generator. Seeing-and-hearing (Xing et al., 2024) propose using ImageBind (Girdhar et al., 2023) as a bridge between visual and audio, leveraging off-the-shelf audio and video generators for multimodal generation. SonicVisionLM (Xie et al., 2024b) converts video-to-audio generation into text-to-audio generation by utilizing a large language model (Chen et al., 2023) to derive video captions for audio generation. Due to the unavailable source codes and non-publicly datasets in SonicVisionLM (Xie et al., 2024b), we tried our best to reproduce it multiple times, and report their best results in our experiments for fair comparison.

**Evaluation Metrics.** We employed several evaluation metrics to assess semantic alignment and audio quality, namely Mean KL Divergence (MKL) (Iashin and Rahtu, 2021), CLIP similarity, and Frechet Distance (FID) (Heusel et al., 2017), following the methodology of previous studies (Luo et al., 2023; Wang et al., 2024a; Xing et al., 2024). MKL measures paired sample-level similarity by calculating the mean KL-divergence across all classes in the test set. CLIP Score compares the similarity between the input video and the generated audio embeddings in the same representation space. For this, we employed Wav2CLIP (Wu et al., 2022) as the audio encoder and CLIP (Radford et al., 2021) as the video encoder, as done in previous works (Wang et al., 2024a; Sheffer and Adi, 2023). FID assesses the distribution similarity to evaluate the fidelity of the generated audio.

For the temporal synchronization, we follow Du et al. (Du et al., 2023; Xie et al., 2024b) and adopt onset detection accuracy (Onset Acc) and onset detection average precision (Onset AP) to evaluate the generated audios, using the onset ground truth from the datasets. However, we identify certain limitations with onset metrics. Firstly, they concentrate on the onset of sound effects while overlooking the persistence of sounds and temporal changes. Therefore, following Du et al. (2023), we also compute the mean absolute error of the audio energy. Secondly, the onset is obtained by setting the threshold of audio amplitude which may lead to inaccuracies. So we follow Yariv et al. (2024) to calculate AV-Align as a supplement.

Table 1: Quantitative evaluation in terms of semantic alignment and audio quality. Specifically, FoleyCrafter achieves state-of-the-art performance with Mean KL Divergence (MKL) (Iashin and Rahtu, 2021), CLIP (Wu et al., 2022) and FID (Heusel et al., 2017) on standard benchmarks, *i.e.,* VGGSound (Chen et al., 2020a) and AVSync15 (Zhang et al., 2024). * denotes reproduction results.

| VGGSound (Chen et al., 2020a) | MKL↓ | CLIP↑ | FID↓ |
|---|---|---|---|
| SpecVQGAN (Iashin and Rahtu, 2021) | 4.337±0.001 | 5.079±0.023 | 65.37±0.01 |
| Diff-Foley (Luo et al., 2023) | 3.318±0.011 | 9.172±0.110 | 29.03±0.61 |
| V2A-Mapper (Wang et al., 2024a) | 2.654 | 9.720 | 24.16 |
| Seeing and Hearing (Xing et al., 2024) | 2.619±0.018 | 2.033±0.147 | 32.99±0.19 |
| SonicVisionLM* (Xie et al., 2024b) | 2.683±0.013 | 9.021±0.187 | 24.42±0.18 |
| FoleyCrafter Timestamp (ours) | 2.612±0.021 | 10.61±0.201 | **19.89**±0.12 |
| FoleyCrafter Energy (ours) | **2.588**±0.019 | **10.63**±0.311 | 20.92±0.12 |
| AVSync15 (Zhang et al., 2024) | MKL ↓ | CLIP ↑ | FID ↓ |
| SpecVQGAN (Iashin and Rahtu, 2021) | 5.339±0.077 | 6.610±0.014 | 114.44±1.31 |
| Diff-Foley (Luo et al., 2023) | 1.963±0.007 | 10.38±0.008 | 65.77±0.01 |
| Seeing and Hearing (Xing et al., 2024) | 2.532±0.021 | 2.098±0.188 | 65.11±1.32 |
| SonicVisionLM* (Xie et al., 2024b) | 2.842±0.023 | 9.236±0.211 | 66.44±1.21 |
| FoleyCrafter Timestamp (ours) | 1.743±0.012 | **11.67**±0.156 | 44.79±1.66 |
| FoleyCrafter Energy (ours) | **1.719**±0.009 | 11.37±0.189 | **42.40**±1.84 |

Table 2: Quantitative evaluation in terms of temporal synchronization. We report onset detection accuracy (Onset ACC), average precision (Onset AP) (Comunità et al., 2024; Xie et al., 2024b), AV-align (Yariv et al., 2024) and Energy MAE (Du et al., 2023) for the generated audios on AVSync (Zhang et al., 2024), which provides onset timestamp labels for assessment, following previous studies (Luo et al., 2023; Xie et al., 2024b). We report the results with error bars calculated from ten times of evaluation with random seeds.

| Method | Onset ACC ↑ | Onset AP ↑ | AV-Align ↑ | Energy MAE↓ |
|---|---|---|---|---|
| SpecVQGAN (Iashin and Rahtu, 2021) | 16.81±2.35 | 64.64±0.72 | 12.42±0.45 | 34.35±0.19 |
| Diff-Foley (Luo et al., 2023) | 21.18±0.08 | 66.55±0.10 | 18.64±0.38 | 41.43±0.11 |
| Seeing and Hearing (Xing et al., 2024) | 20.95±0.87 | 60.33±0.56 | 13.98±0.55 | 38.33±2.42 |
| SonicVisionLM* (Xie et al., 2024b) | **28.89**±1.56 | 68.31±0.94 | 22.06±0.69 | 32.67±0.31 |
| FoleyCrafter Timestamp (ours) | 27.46±2.54 | 69.32±1.03 | 21.93±0.42 | 33.10±0.24 |
| FoleyCrafter Energy (ours) | 24.23±2.60 | **69.91**±0.73 | **22.90**±0.64 | **31.82**±0.45 |

## 4.2 COMPARISON WITH STATE-OF-THE-ART

**Quantitative Comparison.** We present a quantitative comparison of semantic alignment and audio quality on both the VGGSound (Chen et al., 2020a) and AVSync15 (Zhang et al., 2024) datasets, as shown in Table 1. The VGGSound dataset consists of 15,446 videos sourced from YouTube, encompassing a wide range of genres. The results indicate that FoleyCrafter achieves superior semantic alignment with visual conditions and provides higher audio fidelity. Previous approaches encounter difficulties in capturing detailed information from the video due to suboptimal extracted conditions, resulting in limited video alignment. In contrast, FoleyCrafter introduces a semantic adapter that utilizes parallel cross-attention layers to directly integrate video features into the text-to-audio generator, ensuring better alignment with finer details in the videos. Furthermore, we report the results for temporal synchronization on the AVSync15 dataset (Zhang et al., 2024), as displayed in Table 2. The AVSync15 dataset is a carefully curated collection of video-audio pairs with strong video-audio alignment and onset detection labels. This makes it a reliable benchmark for evaluating synchronization. Energy offers more detailed temporal information in audio generation, thereby outperforming other methods based on onset or timestamps (Xie et al., 2024b) Moreover, energy can be simply calculated from mel-spectrum and requires no additional manual annotation.

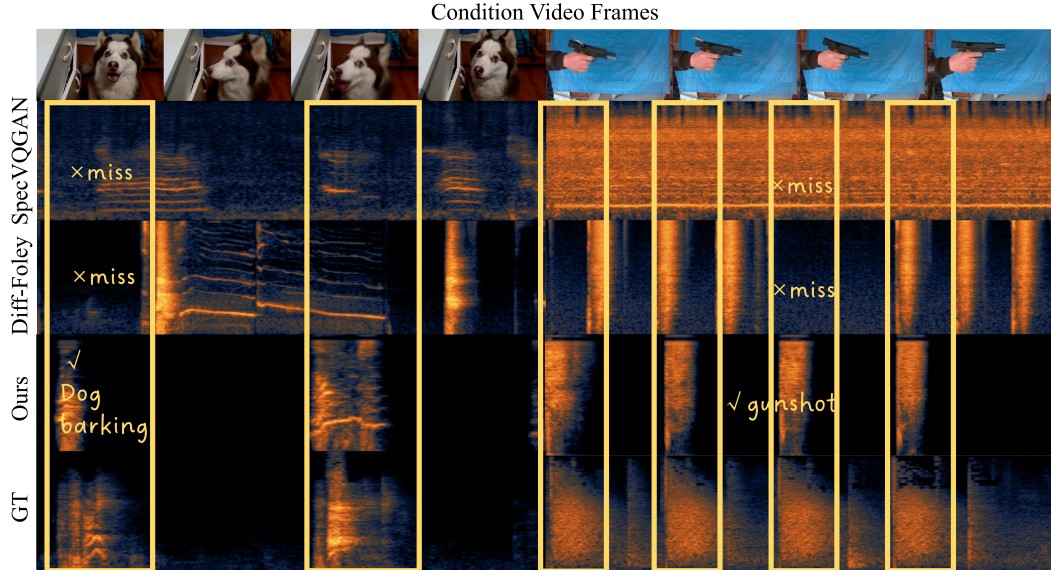

Figure 5: **Qualitative comparison.** As shown in the first case, both SpecVQGAN and Diff-Foley fail to capture the onset of the gunshot sound. In contrast, FoleyCrafter generates the gunshot sound synchronized with the video, showcasing its superior temporal alignment capability.

Table 4: Ablation on temporal adapter.

| Method | Onset Acc↑ | Onset AP↑ | AV-Align↑ | Energy MAE↓ |
|---|---|---|---|---|
| w/o temporal | **26.65** | 63.20 | 21.46 | 36.12 |
| w temporal | 24.23 | **69.91** | **22.90** | **31.82** |

**Qualitative Comparison.** We provide the visualization of generated audio for qualitative comparison on the AVSync15 (Zhang et al., 2024) in Figure 5. It can be observed that FoleyCrafter generates sound at the most accurate time aligned with visual cues, closely resembling the pattern of the ground truth audio. However, SpecVQGAN (Iashin and Rahtu, 2021) tends to introduce more noise, while Diff-Foley (Luo et al., 2023) often generates more or fewer sound events compared to the ground truth. We provide more results in the Appendix.

One notable advantage of FoleyCrafter is its compatibility with text prompts, allowing for more controllable Foley. We present visualization results of audio generation conditioned on both a video and a text prompt in Figure 6. For instance, when the text prompt describes "high pitch," the corresponding value for high-frequency increases compared to when the prompt describes "low pitch." Moreover, FoleyCrafter can also be utilized with negative prompts to prevent the generation of unwanted sounds. In the third case shown in Figure 6, the visual cues depict a horse running on the beach. By setting the negative prompt as "wind and noise" during inference, the generated audio successfully removes the sound of wind and other environmental noise, resulting in a clear sound of hooves. We provide more comparison results in the Appendix.

## 4.3 ABLATION STUDY

We conduct ablation studies to validate the effectiveness of semantic adapter and temporal adapter. For semantic adapter, we compare the audio-visual relevance of generated samples using different methods of video information injection. We consider several baselines for comparison. First, we use a captioner model that

Table 3: Ablation on semantic adapter.

| Method | MKL↓ | CLIP Score↑ | FID↓ |
|---|---|---|---|
| Image embedding | 5.383 | 2.133 | 99.77 |
| Image embedding* | 5.821 | 2.778 | 95.78 |
| Text captioner | 2.331 | 9.177 | 67.40 |
| Semantic adapter (Ours) | **1.719** | **11.37** | **42.40** |

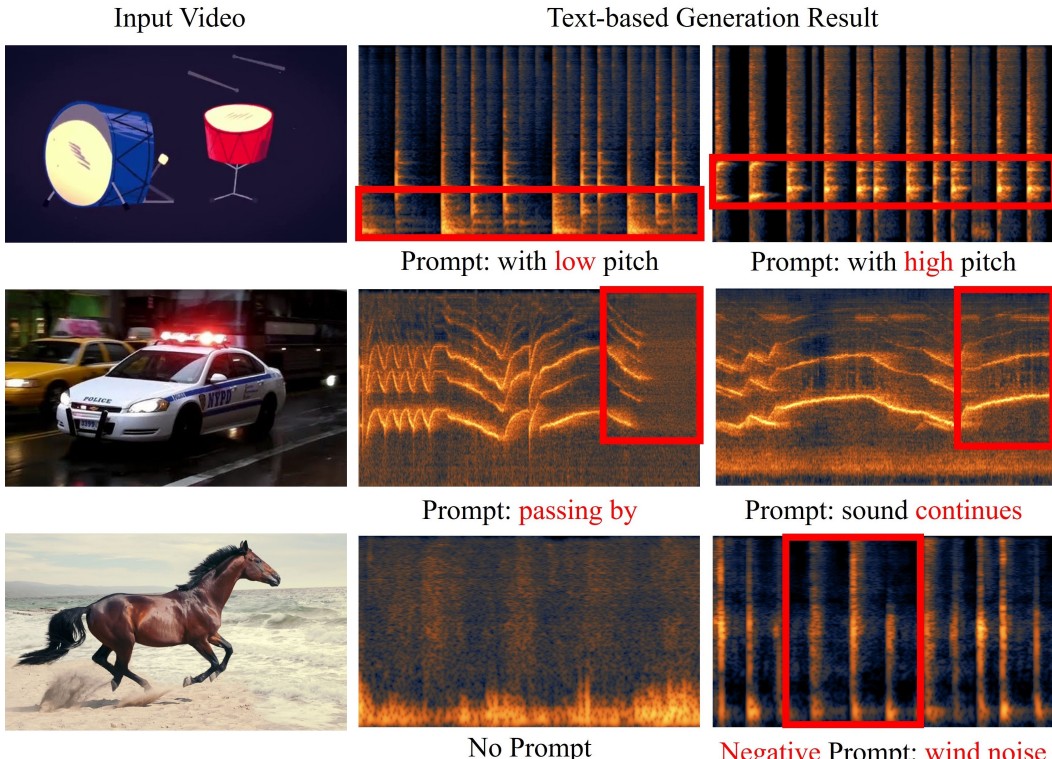

Input Video          Text-based Generation Result

Prompt: with low pitch      Prompt: with high pitch

Prompt: passing by      Prompt: sound continues

No Prompt      Negative Prompt: wind noise

Figure 6: **Video-to-audio generation with text prompts.** FoleyCrafter enhances controllability in video-to-audio generation through text prompts. In the first case, providing a prompt for "high pitch" increases the corresponding value for the drum video. In the third case, a negative prompt like "wind noise" can be used during inference to prevent the generation of wind noise for the video.

utilizes a video-text captioning model (Achiam et al., 2023) to generate text descriptions as inputs to the text-to-audio generator. Second, we directly feed the visual embedding into cross-attention as the text prompt embedding, without any training. Third, we fine-tune the cross-attention module to adapt it to the visual embedding. As shown in Table 3, 'Image embedding' denotes using image clip embedding instead of text embedding as the input of the original cross attention blocks. Besides, 'Image embedding*' denotes the results with further fine-tuning cross-attention blocks. We observed that the caption-based method struggles to capture all the details in the video, resulting in sub-optimal generation results with visual captioning. Using the visual embedding with or without fine-tuning UNet both fail to generate relevant audio for the input video. We attribute this to the significant distortion of the original text-to-audio framework when incorporating visual information.

For temporal adapter, we compare the temporal synchronization performance of FoleyCrafter with and without the module. The results in Table 4 demonstrate that the absence of the temporal adapter leads to a noticeable decline. This decline can be attributed to the fact that the semantic adapter is only capable of capturing video-level semantic information without accurate synchronization features. As a result, it tends to synthesize relevant sounds but with random onset timestamps, leading to a lack of precise temporal alignment.

## 5 CONCLUSION

In this paper, we introduce FoleyCrafter for adding sound effects to silent videos. Unlike existing methods that either train a video-to-audio generator from scratch or use video-to-text translation followed by text-to-audio generation, FoleyCrafter is a pluggable module seamlessly integrated into a text-to-audio generator. This integration ensures high-quality audio generation while synchronizing with the video content. FoleyCrafter leverages two key components, namely semantic adapter for semantic alignment and temporal adapter for temporal synchronization. Extensive experiments on standard benchmarks demonstrate the effectiveness of FoleyCrafter.

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

**Overview.** The appendix includes the following sections:

- **Appendix A.** Limitations and broader impact.
- **Appendix B.2.** Details of training datasets.
- **Appendix B.3.** Details of evaluation.
- **Appendix B.4.** Details of the user study.
- **Appendix C.** More qualitative results.

**Video Result.** We also present video results in a separate supplementary file sourced from Sora.

## A  LIMITATIONS AND BROADER IMPACT

### A.1  LIMITATIONS.

Firstly, although the inclusion of the temporal adapter enhances the synchronization between the generated audio and the input video, its performance can be ultimately limited by the capabilities of the signal estimator. Second, the effectiveness of the signal estimator is contingent upon the availability of strong and relevant training data. When dealing with more complex visual scenes or extremely long videos, predicting the temporal signals for accurate synchronization becomes challenging due to the scarcity of training data in those specific contexts. The visualization results for failure cases are presented in Figure 7.

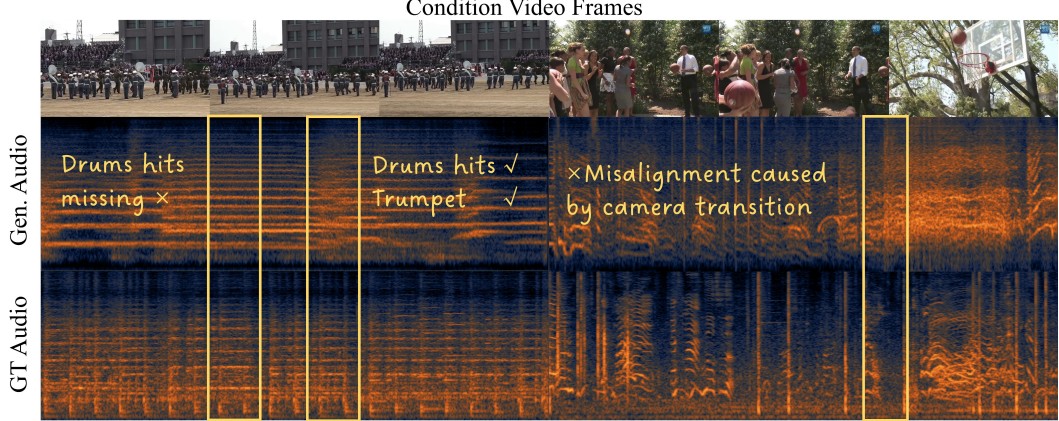

Figure 7: **Failure cases of temporal misalignment.** Left: When dealing with a video scene that contains multiple sounds, such as trumpets and drums, the predicted temporal signals do not accurately reflect the arrangement of each sound, resulting in missing audio. Right: Long videos often contain camera cuts, making it difficult for the temporal estimator to accurately predict the correct temporal signals, which leads to temporal misalignment.

### A.2  BROADER IMPACT.

FoleyCrafter facilitates text-based video-to-audio generation, enabling the generation of sound effects for silent videos and providing control through user prompts. However, it is crucial to acknowledge the potential misuse of such technology for generating fake content on video platforms or social platforms. Users and researchers are strongly advised to exercise caution and carefully screen the use of such technologies to ensure responsible and ethical application.

## B  DETAILS OF EXPERIMENTS

### B.1  DIFFERENCE BETWEEN TIMESTAMP AND ENERGY

Timestamp is a binary mask that indicates the presence or absence of sound effects at each sample point in the audio. Following Xie et al. (2024b); Comunità et al. (2024), we firstly predict the

probability for the sound appearing and then use a threshold to convert it to a binary mask. Audio Energy can be calculated from mel-spectrum Du et al. (2023); Jeong et al. (2024). We use the code from Zhang et al. (2023b) to obtain normalized energy ground truth in training dataset. In practice, our estimator outputs timestamps or energy values with the same length as the input video frames. We then interpolate these to match the length of the audio latent representation.

## B.2 DETAILS OF TRAINING DATASET

FoleyCrafter consist of two key components: semantic adapter and temporal adapter which are trained separately. For the training of semantic adapter we use VGGSound (Chen et al., 2020a) as the training set. VGGSound is an audio-visual dataset containing approximately 199,176 videos sourced from YouTube with annotated label classes indicating the video content. We add the prefix 'The sound of' to the label to form the prompt for generation. We train the timestamp and energy estimator on the AVSync15 (Zhang et al., 2024). AVSync15 is a carefully curated dataset from the VGGSound Sync (Chen et al., 2021) dataset, which contains 1500 strongly correlated audio-visual pairs, making it a high-quality dataset for temporal synchronization. For both timestamp and energy estimator, we train them for 30 epochs. When training the timestamp-based temporal adapter, we need the ground truth timestamp lables for sound event. So we train it on AudioSet Strong (Hershey et al., 2021) which contains 103,463 videos with the audio and the corresponding timestamp labels. For the training of energy-based adapter, we also use the VGGSound Chen et al. (2020a) as we can simply obtain ground truth energy from mel-spectrum.

## B.3 DETAILS OF EVALUATION

We compare timestamp-based and energy-based FoleyCrafter with state-of-the-art methods. For the predicted timestamp, we follow Xie et al. (2024b); Comunità et al. (2024) to use a threshold of 0.5 to get binary timestamp mask. The timestamp and energy condition are is interpolated to the same length as the audio latent. Then they are fed to the ControlNet Zhang et al. (2023a) with the weight of 0.3. For video-to-audio generation, we set the semantic adapter weight to 1.0 and and leave the text prompt empty. This ensures that the semantic information of generated audio is entirely derived from visual.

## B.4 DETAILS OF USER STUDY

To further obtain subjective evaluation results, we conduct a user study. We randomly selected the VGGSound test results generated by different methods for the questionnaire. A total of 20 participants answered our questions. As shown in Figure 8, each question contains audios generated by two methods, one is our method and the other is the baseline e.g. SpecVQGAN (Iashin and Rahtu, 2021) Diff-Foley (Luo et al., 2023) and V2A-Mapper (Wang et al., 2024a). We ask participants to select the one that has better semantic alignment, temporal alignment, and generation quality. Then the preference score can be calculated as

$$Score = \frac{S}{A} \tag{12}$$

where $S$ is the number of times the method has been selected and $A$ is the appearance times of that method. A higher score means the greater performance of FoleyCrafter. Results can be found at Table 5. FoleyCrafter is preferred by users in all three metrics.

## C MORE QUALITATIVE RESULTS

**Foley Generation for Generated Videos.** FoleyCrafter is an effective Foley generation tool which can also be used for movie and generated video. Herein, we take the Sora video as example and provide the audio results generated by FoleyCrafter. In the foley process, semantic adapter can directly utilize the rich visual information, which helps FoleyCrafter generate appropriate sound effects for the visual subjects and environment shown in the generated videos.

**Text-based video to audio generation.** FoleyCrafter achieve text-based video-to-audio generation through parallel cross-attention in semantic adapter. Benefiting from this module, FoleyCrafter

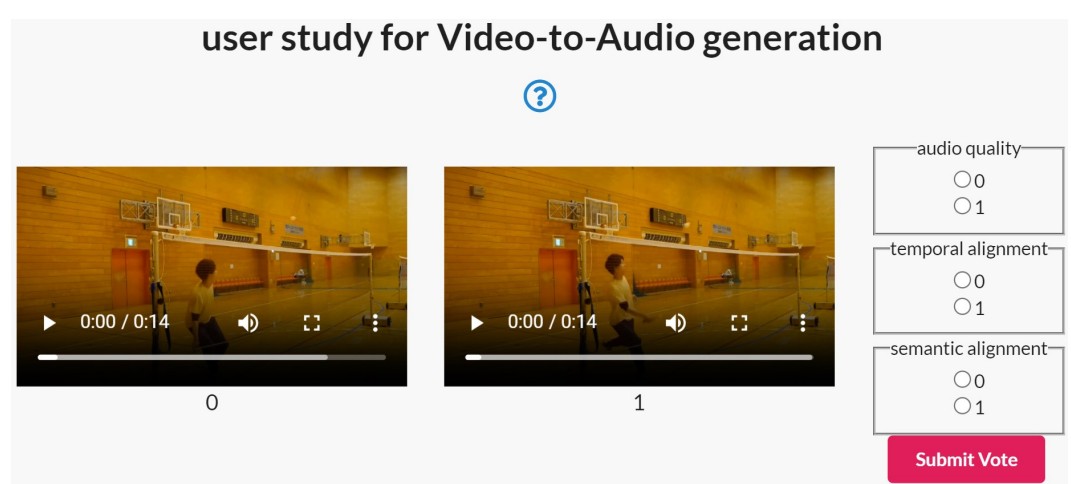

Figure 8: **Screenshot of User Study.**

Table 5: **User study.** We evaluated the performance of three metrics of different models i.e. semantic and temporal alignment and generation quality.

| Method | Semantic | Temporal | Quality |
|---|---|---|---|
| SpecVQGAN | 20.29 | 21.74 | 20.29 |
| Diff-Foley | 20.59 | 29.41 | 27.94 |
| V2A-Mapper | 44.00 | 44.00 | 42.67 |
| FoleyCrafter (ours) | **71.23** | **67.92** | **69.34** |

can utilize both visual information and text prompts to generate audio. Extra text-based video-to-audio generation results are illustrated in Figure 9 and attached in a separate supplementary file.

**Temporal Synchronization Comparison.** The temporal controller enhances the temporal alignment in generated audios with visual cues. To show the synchronization ability of FoleyCrafter, we show more intuitive comparison results between FoleyCrafter and other methods as shown in Figure 10. Video results are also provided in a separate supplementary file.

**Video to Audio generation on various genres.** FoleyCrafter can generate audio for a wide variety of videos. In the supplementary file, we provide generated audio-visual pairs from the VGGSound test cases. The type of video contains realistic video, games, and animation. The main visual objects in the video are people, animals, musical instruments, etc. It fully demonstrates the excellent video-to-audio generation capabilities of FoleyCrafter.

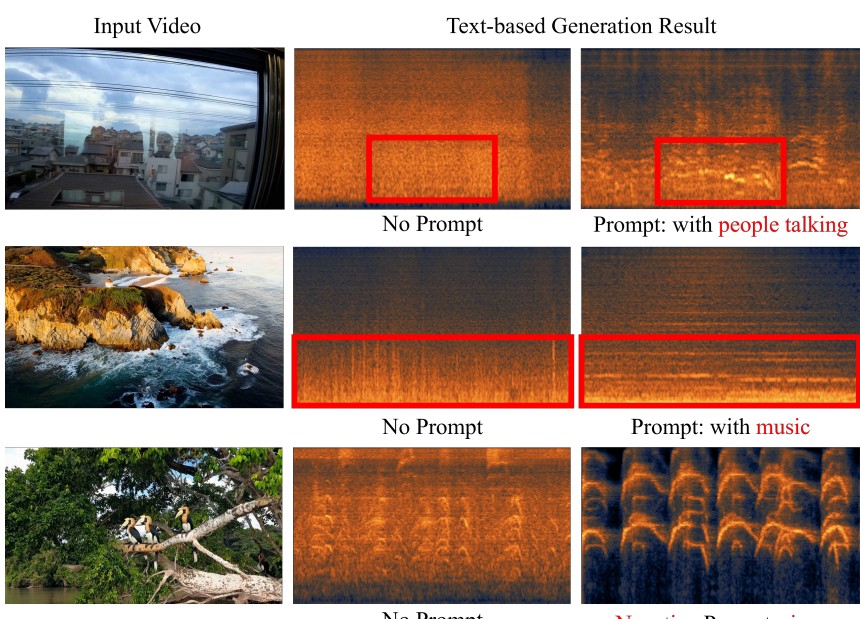

Figure 9: **Extra examples on text-based video to audio generation.**

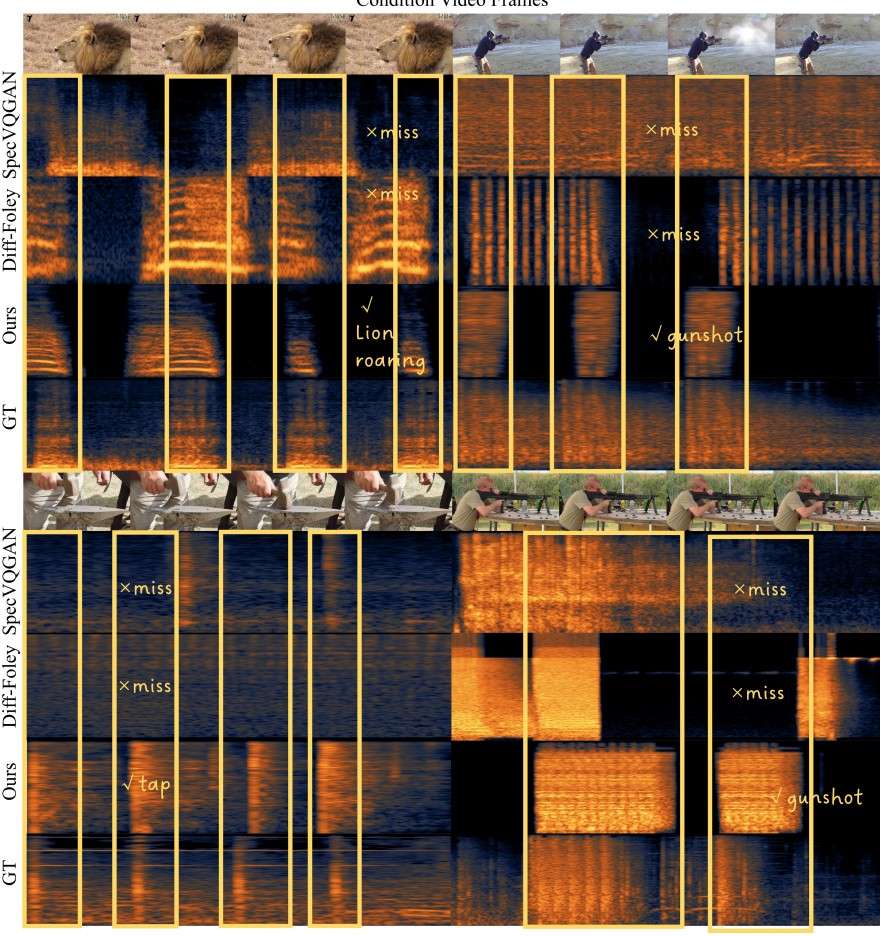

Figure 10: **Extra examples on temporal alignment comparison.**