# OpenReview forum: "FOLEYCRAFTER: BRING SILENT VIDEOS TO LIFE WITH LIFELIKE AND SYNCHRONIZED SOUNDS"
_ICLR.cc/2025/Conference — Submitted to ICLR 2025_

### Official Review · Reviewer_p9QH · 2024-10-30

**Soundness:** 2
**Presentation:** 1
**Contribution:** 2
**Rating:** 3
**Confidence:** 4

**Summary:**

In this paper, the authors proposed a framework called FoleyCrafter to synthesize high-qulity audio with text prompt, which contains two key components as follows:
1.Semantic adapter condition generated audio conditioned on video features, rendering more semantically relevance.
2.Temporal adapter estimates time signals, synchronizing with audio.
The authors carried experiments on two datasets and achieved better performance compared with current powerful models.

**Strengths:**

1.Originality: The authors proposed two adapters to improve the audio synthesis. However, the structure inside originates from other works.
2.Quality: Although the method proposed is effective compared to others, it lacks rigorous mathematical proof.
3.Clarity: Semantic adapter has not been clarified clearly, especially the cross-attention component.
4.Significance: The significance of the method is relatively high comparing to existing methods. However, parameters to be trained is relatively high compared to others.

**Weaknesses:**

1.Lack of Innovation: In this article, there are two key components. However, the semantic adapter is derived from the IP-adapter[1], while the temporal adapter originates from ControlNet[2]. This article lacks substantial original contributions.
2.Inference Latency Concerns: In the articles mentioned above, the authors only add a single adapter to the original model. However, in this article, the proposed method includes two separate adapters, which may result in higher inference latency, potentially impeding efficiency and scalability.
3.Insufficient Analysis of Text Prompts: In this article, there are text prompts and video prompts for audio generation. However, The authors provide only a qualitative description of the text prompt's capabilities, without comparing it to other models.

[1] Hu Ye, Jun Zhang, Sibo Liu, Xiao Han, and Wei Yang. Ip-adapter: Text compatible image prompt adapter for text-to-image diffusion models. arXiv preprint arXiv:2308.06721, 2023.
[2] Lvmin Zhang, Anyi Rao, and Maneesh Agrawala. Adding conditional control to text-to-image diffusion models. In Proceedings of the IEEE/CVF International Conference on Computer Vision, pages 3836–3847, 2023a.

**Questions:**

No

---

> ### Author Response · Authors · 2024-11-20
>
> We sincerely appreciate your review and addess the major concerns below.
>
> **W1. Limited innovations**
>
> We respectfully disagree with the assessment of limited innovation. FoleyCrafter introduces several technical breakthroughs:
>
> 1. Novel End-to-End Approach:
>   - Previous methods either use video-to-text-to-audio pipelines (Xing et al., 2024; Wang et al., 2024; Xie et al., 2024b) compromising video alignment, or train from scratch on noisy datasets (Iashin and Rahtu, 2021; Luo et al., 2023) sacrificing audio quality.
>   - FoleyCrafter is the first to generate high-quality audio directly from video while maintaining strong temporal alignment.
> 2. We attribute the breakthroughs of FoleyCrafter by the following model designs:
> - Semantic Adapter: While sharing architectural similarities with IP-Adapter, we pioneer its application for cross-modality conditioning from video to audio which differs from image-to-image in IP-Adapter. As discussed in L263-269, we develop a specialized video encoder to capture visual cues for audio generation. Our novel random dropping strategy enables both effective visual guidance and text controllability, providing a new paradigm for video-guided audio synthesis.
> - Temporal Adapter: Unlike traditional ControlNet which performs spatially aligned feature residual addition from image to image, we address the unique challenge of video-audio temporal alignment through two novel approaches:
>    - Timestamp event mask-based method
>    - Energy map-based control that eliminates the need for video labels, enabling training on larger-scale datasets.
>
> These adapters make FoleyCrafter the first to employ plug-and-play modules to achieve high-quality, video-aligned audio generation **directly from visual content**.
>
> 3. These innovations deliver substantial improvements over the strongest baselines:
> - 14.33% improvement in audio quality (FID scores).
> - 10.6% Better temporal alignment performance across multiple metrics.
> - Unique flexibility supporting both video-only and video-text inputs.
> Our work bridges the gap between high-quality audio synthesis and precise video alignment, establishing a new paradigm for neural foley sound generation.
>
> **W2.  Inference latency and trainable parameters.**
>
> FoleyCrafter demonstrates superior inference time. We evaluate the 10-second audio generation time for different methods using the same video frame rate on the same device, and present the results below. For implementation, both the semantic adapter and temporal adapter can be parallelized with the UNet during the sampling process, resulting in fast inference times.
>
> | Method             | Inference Time | Trainable Parameters |
> |--------------------|----------------|----------------------|
> | SpecVQGAN (ResNet) | 4.8s          | 379M                 |
> | Diff-Foley         | 2.7s          | 859M                 |
> | Seeing-and-Hearing | 22.02s        | -                    |
> | SonicVisionLM      | 3.7s          | 364M                 |
> | Ours               | 3.1s          | 415M                 |
>
> The total trainable parameters include two adapters for 415M and temporal estimator for 31M which is a lot smaller than Diff-foley with 859M and comparable with SpecVQGAN and SonicVisionLM with approximately 400M.
>
> **W3. Analysis of text prompts.**
>
> We would like to clarify that FoleyCrafter is primarily designed for video-to-audio generation, without requiring text input. While most existing video-to-audio models are limited to visual inputs, our semantic adapter uniquely enables optional text-based control. We demonstrate this additional capability qualitatively in Figure 6 of the main paper.
>
> In response to the reviewer's concern, we conducted additional comparison of text-based video-to-audio generation results with those of other methods. As described in the main paper, here we also use wav2clip (Wu et al., 2022) to calculate the embedding similarity between audio embeddings, text embeddings and visual embeddings. We conduct the evaluation on AVSync15 (Zhang et al., 2024) and use the "The sound of [label]" as prompt.
>
> As shown in the table below, when both text prompts and visual information are provided, FoleyCrafter achieves the best performance on the text clip score, indicating the **best alignment with the prompt**.  These results demonstrate FoleyCrafter's **flexible ability to condition generation on both text and video** inputs according to users' intents.
>
> | Method                        | CLIP-Visual | CLIP-Text |
> |-------------------------------|-------------|-----------|
> | SpecVQGAN (ResNet)            | 6.610       | 17.92     |
> | Diff-Foley                    | 10.38       | 17.32     |
> | Seeing-and-Hearing            | 2.098       | 17.45     |
> | SonicVisionLM                 | 9.236       | 17.21     |
> | FoleyCrafter (V2A)           | **11.67**       | 17.98     |
> | FoleyCrafter (text-based V2A) | 11.21       | **18.07**     |

---

> > ### Author Response · Authors · 2024-11-25
> > **Response to Reviewer p9QH (Part 2/2)**
> >
> > **[About Potential Failure Cases]**
> >
> > - We have indeed discussed potential failure cases in the limitations section (L716-L722) of the appendix. Specifically, when the visual scene becomes highly complex or the video is exceptionally long, synchronization accuracy can be constrained by the performance of the audio signal estimation and the quality of the training data.
> >
> > - Based on your suggestions, we have also visualized and included some failure cases in the revised version. Please refer to the uploaded revision for our visualization analysis. To address these challenges in temporal alignment, we plan to focus on constructing highly visual-audio-aligned datasets and advancing model design in future work.
> >
> >
> > **[About Details of Subjective Evaluations]**
> >
> > Thank you for pointing out this issue. In addition to the details provided in Section B.3 of the appendix, we will incorporate the following clarifications in the final version of the paper, as per your suggestions.
> >
> > We conducted a user study involving **20 participants** who rated 40 randomly selected video-audio samples, following V2A-Mapper. Specifically, the participants were either practitioners in audio generation and multimedia or PhD students specializing in Artificial Intelligence. To ensure unbiased feedback, all results were presented to the participants anonymously.
> >
> > Evaluating audio quality, temporal alignment, and semantic alignment across samples from different models requires significant focus from participants. To reduce cognitive load and avoid random ratings, we opted for **pairwise comparisons** instead of Mean Opinion Scores or Meaningful Difference Scores, as used in V2A-Mapper. As shown in Figure 3 of the appendix, pairwise comparisons simplify the evaluation process by asking participants to compare two results at a time, which improves the reliability of their judgments and ensures higher utilization of user study votes. Such a pairwise comparison design is also widely used in the fields of large language models (LLMs) [1,2] and computer vision [3,4]. Specifically, in each trial, participants were presented with two results: one generated by FoleyCrafter and the other by a randomly selected baseline.
> >
> > To further ensure the quality and consistency of the evaluations, we designed the study to present the same questions multiple times throughout the process. This repetition helped us verify participants’ attentiveness and identify any inconsistencies in their responses. Inconsistent scores for the same question from the same participant were treated as unreliable, allowing us to maintain the integrity of the results.
> >
> > [1] Liu, Yinhong, et al. "Aligning with human judgement: The role of pairwise preference in large language model evaluators." arXiv preprint arXiv:2403.16950 (2024).
> >
> > [2] Liusie, Adian, et al. "Efficient LLM Comparative Assessment: a Product of Experts Framework for Pairwise Comparisons." arXiv preprint arXiv:2405.05894 (2024).
> >
> > [3] Li, Shufan, et al. "Aligning diffusion models by optimizing human utility." arXiv preprint arXiv:2404.04465 (2024).
> >
> > [4] Zeng, Yanhong, et al. "Aggregated contextual transformations for high-resolution image inpainting." IEEE Transactions on Visualization and Computer Graphics 29.7 (2022): 3266-3280.

---

> ### Author Response · Authors · 2024-11-22
> **Please let us know whether we address all the issues**
>
> Dear reviewer,
>
> We have submitted the response to your comments. Please let us know if you have additional questions so that we can address them during the discussion period. We hope that you can consider the raising score after we address all the issues.
>
> If you still have questions and concerns, please feel free to comment here. We will reply it as soon as possible.
>
> Thank you!

---

> ### Comment · Reviewer_p9QH · 2024-11-24
>
> I appreciate the author’s responses to my concerns. However, I would like to share some remaining concerns that have not yet been fully resolved.
>
> **[About Novelty]**
> The contributions of this work are primarily based on the effective application of existing architectures. While the authors have clarified that the Semantic Adapter and Temporal Adapter represent new applications of existing methods, the work lacks further theoretical analysis or novel architectural design that aligns with the quality of ICLR in the context of video-to-audio generation.
>
> **[About Inference latency and trainable parameters]**
> Could you clarify why Diff-Foley achieves lower inference latency despite having twice the number of parameters? Is it due to the proposed two separate adapters? Additionally, some experimental details are missing. For instance, do FoleyCrafter and baselines employ acceleration techniques for attention, such as flash-attention? What is the exact inference time for each submodule of FoleyCrafter? Addressing these points could help make the results more compelling.
>
> **[About Analysis of text prompts]**
> Thank you for providing the experimental results. The response from the authors has addressed my questions.
>
>
> **[About Potential Failure Cases]**
> Although Figure 6 shows the effectiveness of the proposed method, the paper does not adequately discuss potential failure cases in video to audio generation. It would be beneficial for the authors to address scenarios where the model might underperform, such as wrong temporal information predicted by the temporal estimator that lead to failures, difficulties with extremely long video inputs, or challenges with some special video contents. Understanding these limitations would provide a clearer understanding of the proposed model.
>
> **[About Details of Subjective Evaluations]**
>  The details of the subjective evaluations are missing, i.e., what the demographics are like for the raters in subjective evaluations, whether there are attention checkers, how the results are quality-checked etc. Besides, there is no information about the compensation, or criteria for hiring human subjects for the subjective evaluation.

---

> ### Author Response · Authors · 2024-11-25
> **Response to Reviewer p9QH (Part 1/2)**
>
> Dear reviewer:
>
> Thanks for your comments. Here are our replies for your remaining concerns.
>
> **[About Novelty]**
>
> Thank you for acknowledging our new applications of the Semantic Adapter and Temporal Adapter for video-to-audio generation. We argue that the potential of **developing plug-and-play modules based on off-the-shelf T2A (text-to-audio) models for video-to-audio generation is highly underrated**. High-quality text-audio paired datasets are significantly larger and more reliable compared to video-audio paired datasets, which are often low-quality and inconsistent. We believe that training plug-and-play modules for video-to-audio adaptation based on text-to-audio models is a promising direction to achieve high-quality results while maintaining training efficiency, especially compared to training video-to-audio models from scratch (e.g., SpecVQGAN and Diff-Foley).
>
> To advance the development of such a plug-and-play mechanism:
>
> 1. **We take the first step** by designing a Semantic Adapter **to encode video features for direct attention** by audio generation models. This approach avoids relying on text as an intermediate bridge, as seen in prior works (e.g., Seeing and Hearing, SonicVisionLM, and V2A-Mapper), which often leads to suboptimal semantic alignment.
>
> 2. **We conducted the first extensive study on both timestamp-based and energy-map-based audio signal estimation** to address the spatial misalignment problem between visual and audio modalities when applying ControlNet.
>
> We sincerely hope that these insights and explorations, supported by our extensive experiments and strong results, will inspire future research and drive progress in the video-to-audio community.
>
>
> **[About Inference latency and trainable parameters]**
>
> Thanks for pointing out this issue. We will add the following clarifications to the final version.
>
> - **Inference latency of Diff-Foley.** While Diff-Foley has twice the number of **trainable parameters** (859M for the entire model trained from scratch), it achieves lower latency during inference since its **total model size remains 859M**. While FoleyCrafter has **fewer trainable parameters (415M for adapters)**, it has about 1.2B (additional 890M for the frozen UNet) model parameters during inference, leading to slightly higher inference latency. It is important to note that inference latency can also be influenced by various design factors (e.g., network architecture, downsampling factors, feature extraction, etc.). Nonetheless, by comparing the overall inference latency and trainable parameters, we demonstrate that FoleyCrafter achieves competitive inference latency while offering the added advantage of high training efficiency, thanks to its plug-and-play framework.
>
> - **Comparison settings**. We have checked and ensured that all the models (including FoleyCrafter) were tested using their native implementations, without employing acceleration techniques such as flash attention, to ensure a fair comparison.
>
> - **Exact inference time for each submodule**. The exact inference time for each submodule of FoleyCrafter is as follows: **Semantic Adapter** (**0.58** seconds for visual encoding), **Temporal Adapter** (**0.12** seconds for audio signal estimation), **UNet inference** (**1.59** seconds for complete diffusion sampling), and **IO** (**0.84** seconds for reading video frames), resulting in **a total inference time of 3.1 seconds**. We recognize that FoleyCrafter has potential for further optimization in terms of inference latency, which we plan to address in future work.

---

> ### Author Response · Authors · 2024-11-27
> **Further Concerns or Questions?**
>
> We have provided additional details and clarifications in response to your remaining concerns, and we would like to know if there are any further questions or issues we can address. We are committed to engaging fully and will reply as promptly as possible.
>
> Thank you once again for your insightful and constructive feedback, which we believe has been instrumental in strengthening our work. We sincerely hope the improvements we have made address your key concerns, highlight the contributions of our paper, and contribute positively to the development of this community. We kindly ask you to reconsider your evaluation in light of these changes, and we remain happy to further address any additional concerns you may have.

---

> ### Author Response · Authors · 2024-11-28
> **We are looking forward to your feedback.**
>
> Dear Reviewer p9QH,
>
> As the discussion deadline approaches, we would like to know whether we have addressed your remaining concerns.
>
> We highly value your feedback and have made additional clarifications and necessary revisions in the manuscript, highlighted in $\color{red}{red}$. Your time and feedback are greatly appreciated, and we look forward to your response.

---

### Official Review · Reviewer_Qwiq · 2024-10-30

**Soundness:** 3
**Presentation:** 3
**Contribution:** 3
**Rating:** 6
**Confidence:** 1

**Summary:**

This paper introduces FoleyCrafter, a framework designed for automatically generating realistic and synchronized sound effects for silent videos. FoleyCrafter leverages a pre-trained text-to-audio model, incorporating a “semantic adapter” and “temporal adapter” to ensure that the generated audio is semantically aligned with video content and precisely synchronized over time. Additionally, it supports customizable audio generation through text prompts. The primary contributions include: 1) presenting a novel neural Foley framework for high-quality, video-aligned sound generation, 2) designing semantic and temporal adapters to improve audio-video alignment, and 3) achieving state-of-the-art performance on benchmarks through comprehensive quantitative and qualitative evaluations.

**Strengths:**

1.	Originality: This paper introduces an innovative framework, FoleyCrafter, which stands out in the field of sound generation for silent videos. By combining a pre-trained text-to-audio model with novel adapter designs (semantic and temporal adapters), it effectively addresses the limitations of existing methods in terms of audio quality and video synchronization, showcasing unique and original thinking.
2.	Quality: The paper demonstrates high research quality through comprehensive experimental design and implementation. It includes extensive quantitative and qualitative experiments, validating the effectiveness of FoleyCrafter on standard benchmark datasets. The results show that this method surpasses several state-of-the-art approaches in both audio quality and synchronization performance. Additionally, the availability of code and models facilitates future replication and research.
3.	Clarity: The paper is well-structured, with clear explanations of concepts and model design, allowing readers to easily understand how FoleyCrafter operates. The figures and results in the experimental section are also well-presented, enabling readers to intuitively grasp the method’s performance and advantages.
4.	Significance: FoleyCrafter holds substantial application potential in the field of video-to-audio generation. This approach not only enhances the realism and synchronization of sound effects but also offers controllability and diversity through text-based prompts. Such innovations have broad applicability in multimedia production, including film and gaming, and further advance cross-modal generation technology in the audio-visual domain.

**Weaknesses:**

The paper’s originality appears limited. The whole model system exploits many present models, such as Freesound Project and Auffusion.
Although the part of Quantitative Comparison includes evaluations in terms of semantic alignment, audio quality and temporal synchronization, the comparison of audio generation speed has not been expressed.
The lack of some ablation experiments for Semantic Adapter and Temporal Controller weakens persuasiveness. The Semantic Adapter could be entirely removed to observe the system’s performance without visual semantic information. The Onset Detector and Timestamp-Based Adapter could be individually removed to investigate their roles in temporal alignment and onset detection. In addition, it would be more persuasive if ablation experiments for Parallel Cross-Attention with different λ had been done.

**Questions:**

Refer to Weakness

---

> ### Author Response · Authors · 2024-11-20
>
> We sincerely appreciate your positive feedback regarding our 'unique and original thinking,' 'high research quality with extensive experiments,' 'well-structured' presentation, and 'innovations with substantial application potential.' We address the remaining concerns below.
>
> **W1. Novalty in leveraging T2A models for V2A generation.**
>
> FoleyCrafter is the first to employ plug-and-play modules to achieve high-quality, video-aligned audio generation **directly from visual content**.
> It investigates the off-the-shelf high-quality pre-trained audio generator for video-to-audio (V2A) generation. The pre-trained model exhibits a robust text-to-audio generative capability that we find advantageous for V2A tasks. By utilizing this powerful audio generator, FoleyCafter can produce more realistic and higher-quality audio compared to existing video-to-audio models.
>
> However, generating video-aligned audio with such a well-trained audio generator still remains a challenge. To address this, we propose the semantic adapter and temporal adapter to enable visual relevant and synchronized audio generation. In summary, we investigated how to leverage the existing T2A model and introduced new modules to adapt it for V2A tasks.
>
> **W2. About the generation speed.**
>
> FoleyCrafter has a fast generation speed.
> We evaluate the inference time of existing works using the same video frame rate and on the same computing device, and report the results below.
>
> | Method             | Inference Time | Trainable Parameters |
> |--------------------|----------------|----------------------|
> | SpecVQGAN (ResNet) | 4.8s          | 379M                 |
> | Diff-Foley         | 2.7s          | 859M                 |
> | Seeing-and-Hearing | 22.02s        | -                    |
> | SonicVisionLM      | 3.7s          | 364M                 |
> | Ours               | 3.1s          | 415M                 |
>
> For implementation, both the semantic adapter and temporal adapter can be parallelized with the UNet during the sampling process, resulting in fast inference times.
>
> **W3. Ablation of Semantic Adapter and Temporal Adapter.**
>
> We indeed include ablation experiments for semantic and temporal adapters in Table 3 and Table 4 of the main paper.
>
> We report audio quality and audio-visual relevance with and without the semantic adapter in Table 3 L483-L484. The improvement in MKL and CLIP scores indicates that the semantic adapter effectively integrates detailed visual embeddings for audio generation. Additionally, the lower FID demonstrates that the semantic adapter enhances the utilization of the well-trained audio generator for high-quality video-to-audio generation.
>
> For the temporal adapter, we assess the temporal alignment of audio samples generated with and without it in Table 4 L457-L460. The significant improvement (10.6%) of Onset AP, AV-Align and Energy MAE shows temporal adapter improved the synchronization of FoleyCrafter.
>
> **W4. Ablation of λ in Cross-Attention.**
>
> As described in L240-L269, the semantic adapter utilizes the parallel cross-attention with a variable parameter λ. Here we show the ablation results of different λ. The results indicate that as λ decreases, the MKL, CLIP score, and FID all decline, demonstrating that the semantic adapter is crucial for audio-visual alignment and high-quality generation.
>
> | λ                      | MKL↓       | CLIP↑      | FID↓       |
> |------------------------|------------|------------|------------|
> | λ = 0                  | 6.212      | 3.542      | 103.1      |
> | λ = 0.4                | 2.376      | 8.428      | 61.56      |
> | λ = 0.8                | 1.857      | 9.856      | 49.71      |
> | λ = 1.0 (FoleyCrafter) | **1.719**  | **11.37**  | **42.40**  |

---

> ### Author Response · Authors · 2024-11-25
>
> Dear reviewer,
>
> We have provided comprehensive responses to each concern. Please let us know if you have any additional questions that we can address during the discussion period. We hope that you can consider the raising score after we address all the issues.
>
> If you still have questions and concerns, please feel free to comment here. We will reply it as soon as possible.
>
> Thank you!

---

### Official Review · Reviewer_Bd3G · 2024-11-04

**Soundness:** 2
**Presentation:** 4
**Contribution:** 3
**Rating:** 8
**Confidence:** 4

**Summary:**

This paper presents a new video-to-audio model, featured by the semantic adapter and temporal adapter. The proposed model uses the [Auffusion](https://arxiv.org/abs/2401.01044) model as a baseline, and not only video-audio paired data but also text-audio paired data are used for training its sub-modules for connecting between the visual encoder and Auffusion. The temporal adapter, trained with the BCE loss or MSE loss to estimate the energy map of audio from video, enhances the synchronization between video and audio. The authors conducted both quantitative and qualitative comparisons with previous video-to-audio models to demonstrate that the proposed model outperforms them. They also conducted ablation studies to show that their proposed semantic and temporal adapters are effective.

**Strengths:**

1. Although there is room for improvement in writing style, the paper itself is well-written enough to make readers understand their motivation, the proposed method, and the experimental results.
2. The proposed video-to-audio model is well-designed to address the issue of synchronization between video and audio. There may be other designs for resolving the issue, but they conducted ablation studies to demonstrate that their designed model works well.
3. The authors quantitatively evaluated their model on the commonly used benchmarks and qualitatively analyzed the audio signals generated from the proposed and previous models for comparison. These experimental results show that the proposed model outperforms the previous models.

**Weaknesses:**

- L.365: "We employed several evaluation metrics to assess semantic alignment and audio quality, namely Mean KL Divergence (MKL) (Iashin and Rahtu, 2021), CLIP similarity, and Frechet Distance (FID) (Heusel et al., 2017), following the methodology of previous studies (Luo et al., 2023; Wang et al., 2024; Xing et al., 2024). MKL measures paired sample-level similarity"
  - The application of FID to audio quality evaluations is proposed by [Iashin and Rahtu (2021)](https://www.bmvc2021-virtualconference.com/conference/papers/paper_1213.html), and [Luo et al. (2023)](https://proceedings.neurips.cc/paper_files/paper/2023/hash/98c50f47a37f63477c01558600dd225a-Abstract-Conference.html) followed them. However, [Wang et al. (2024)](https://ojs.aaai.org/index.php/AAAI/article/view/29475) and [Xing et al. (2024)](https://openaccess.thecvf.com/content/CVPR2024/html/Xing_Seeing_and_Hearing_Open-domain_Visual-Audio_Generation_with_Diffusion_Latent_Aligners_CVPR_2024_paper.html) use different metrics, FD ([Liu et al., 2023](https://proceedings.mlr.press/v202/liu23f.html)) and FAD ([Kilgour et al., 2019](https://www.isca-archive.org/interspeech_2019/kilgour19_interspeech.html)). I recommend the authors additionally evaluate their proposed model with these metrics for several reasons. The FID and FAD are calculated from spectrograms and do not consider phase information of audio signals. The FD is based on the PANN network ([Kong et al., 2020](https://ieeexplore.ieee.org/document/9229505)), which takes audio waveforms and achieves better performance in classification tasks than VGGish. Plus, recent papers use FAD or FD more frequently. The evaluation on these metrics will be informative to readers, which means the authors can contribute more to the community.

**Questions:**

I would appreciate the authors' response to my comments in "Weaknesses".

---

> ### Author Response · Authors · 2024-11-20
>
> We sincerely appreciate your review and addess the major concerns below.
>
> **W1. Additional Evaluation in terms of FD and FAD.**
>
> We appreciate the reviewer's suggestion regarding the evaluation metrics. Following this feedback, we conducted comprehensive evaluations using both Fréchet Distance (FD) and Fréchet Audio Distance (FAD) as below. Our results demonstrate that FoleyCrafter consistently outperforms existing methods, achieving state-of-the-art performance across both metrics. Specifically, FoleyCrafter achieves an FD of 20.32 and FAD of 2.69, representing a 19.41% improvement over the strongest baseline.
>
> | Method                   | VGGSound                       |               | AVSync                         |               |
> |--------------------------|-------------------------------|---------------|--------------------------------|---------------|
> |                          | FD↓                           | FAD↓          | FD↓                            | FAD↓          |
> | SpecVQGAN                | 32.08                         | 5.563         | 32.08                          | 11.51         |
> | Diff-Foley               | 29.37                         | 6.199         | 29.37                          | 12.75         |
> | Seeing-and-Hearing       | 33.11                         | 4.325         | 33.11                          | 11.67         |
> | SonicVisionLM            | 21.06                         | 3.338         | 21.06                          | 10.24         |
> | Ours (Timestamp-based)   | 20.33                         | **2.326**     | 20.33                          | 8.045         |
> | Ours (Energy-based)      | **20.32**                    | 2.690         | **20.32**                     | **7.902**     |
>
>
>
> **Improvement of writing style.**
>
> We appreciate the reviewer's detailed suggestions on improving the presentation. We have standardized all table, figure, and citation references, and expanded Section 2 with more comprehensive related works. All changes can be found in our uploaded revision.

---

> > ### Comment · Reviewer_Bd3G · 2024-11-23
> > **Response to Authors**
> >
> > I sincerely appreciate the time and effort the authors spent addressing the reviewers' comments. I feel that all of my concerns have been addressed very effectively. I raised my rating to 8. I recommend the paper.

---

### Author Response · Authors · 2024-11-20

Thank all reviewers for your time and invaluable feedback.

We appreciate the recognition of FoleyCrafter as a **well-designed model** (Bd3G) with **unique and original thinking** (Qwiq) and the acknowledgement of the **comprehensiveness of our experimental results** (Bd3G, Qwiq) which shows it a **effective method** for video-to-audio generation (p9QH).

We address the questions and concerns for each reviewer in the rebuttal sessions individually.   As mentioned in our responses, we will incorporate clarifications in the final version of our paper. Please let us know if there are any questions or further clarifications/discussions.

---

### Meta-Review · Area_Chair_MAV8 · 2024-12-21

**Metareview:**

The paper presents a framework: FoleyCrafter for adding foley sound effects to videos. The key innovations are: i) to enhance an audio latent diffusion model that is conditioned on text to also include video semantics via incorporating video features using cross-attention layers, and ii) temporal synchronization of the audio with the time-varying video content, using either manually-provided time-stamps or using an energy map estimated from mel spectrograms, followed by using ControlNet for synchronization. Experiments on VGGSound and AVSync15 show promising results.

**Additional Comments On Reviewer Discussion:**

The paper is well-written and easy to follow, however received mixed reviews. While the reviewers appreciated the empirical benefits showcased and the substantial potential of the model towards enabling varied applications in video-to-audio generation, there were many concerns brought up, which were debated with the authors during the author-reviewer discussion phase. Mainly, there were three important concerns raised by the reviewers:
1. Lack of originality in the contributions (Qwiq, p9QH)
2. Many ablations studies, performance comparisons, and evaluation metrics being missed (Qwiq, p9QH, Bd3G)
3. Latency in the audio-generation against prior methods.

During the discussion phase, authors provided additional numerical results addressing points 2 and 3. Specifically, authors pointed out  the ablation studies that were already in the paper, comparisons to w/ and w/o text-prompts were provided, inference latency comparisons were provided that showed the latency to be in the same ballpark as prior methods, and comparisons on two additional metrics FD and FAD (requested by Bd3G) were added.

However, the concern regarding novelty remains. As noted by Reviewer p9QH, the contributions of this paper appear incremental to IP-adapter [1] and ControlNet [2]. While, the authors pointed out during the discussion that the paper proposes to use different modalities (video features) as part of their semantic alignment module (as against text features in [1]), AC agrees with the reviewer that there is a lack of any significant novelty made in this incorporation. Further, while the paper uses timestamp information and energy-map of mel spectrograms for estimating the time-varying audio signals, these appear more of heuristic choices. As such, AC agrees with the Reviewers p9QH and Qwiq that, despite the strong empirical performances, unfortunately the paper is short of significant scientific contributions or insights in comparison to prior art, and thus recommends reject.

---

### Decision · Program_Chairs · 2025-01-22

Reject